# New Highly Selective BACE1 Inhibitors and Their Effects on Dendritic Spine Density In Vivo

**DOI:** 10.3390/ijms241512283

**Published:** 2023-07-31

**Authors:** Katrin Pratsch, Chie Unemura, Mana Ito, Stefan F. Lichtenthaler, Naotaka Horiguchi, Jochen Herms

**Affiliations:** 1German Center for Neurodegenerative Diseases (DZNE), 81377 Munich, Germany; katrin.pratsch@med.uni-muenchen.de (K.P.); stefan.lichtenthaler@dzne.de (S.F.L.); 2Munich Cluster for Systems Neurology (SyNergy), 81377 Munich, Germany; 3Center for Neuropathology and Prion Research (ZNP), Faculty of Medicine, LMU Munich, 81377 Munich, Germany; 4Laboratory for Drug Discovery and Disease Research, Shionogi & Co., Ltd., Shionogi Pharmaceutical Research Center, Osaka 561-0825, Japan; chie.unemura@shionogi.co.jp (C.U.); mana.itou@shionogi.co.jp (M.I.); naotaka.horiguchi@shionogi.co.jp (N.H.); 5Neuroproteomics, School of Medicine, Klinikum Rechts der Isar, Technical University of Munich, 81675 Munich, Germany

**Keywords:** Alzheimer’s disease, BACE1, BACE inhibitor, side effects, in vivo two-photon microscopy, dendritic spine plasticity

## Abstract

β-site amyloid precursor protein-cleaving enzyme 1 (BACE1) is considered a therapeutic target to combat Alzheimer’s disease by reducing β-amyloid in the brain. To date, all clinical trials involving the inhibition of BACE1 have been discontinued due to a lack of efficacy or undesirable side effects such as cognitive worsening. The latter could have been the result of the inhibition of BACE at the synapse where it is expressed in high amounts. We have previously shown that prolonged inhibition of BACE interferes with structural synaptic plasticity, most likely due to the diminished processing of the physiological BACE substrate Seizure protein 6 (Sez6) which is exclusively processed by BACE1 and is required for dendritic spine plasticity. Given that BACE1 has significant amino acid similarity with its homolog BACE2, the inhibition of BACE2 may cause some of the side effects, as most BACE inhibitors do not discriminate between the two. In this study, we used newly developed BACE inhibitors that have a different chemotype from previously developed inhibitors and a high selectivity for BACE1 over BACE2. By using longitudinal in vivo two-photon microscopy, we investigated the effect on dendritic spine dynamics of pyramidal layer V neurons in the somatosensory cortex in mice treated with highly selective BACE1 inhibitors. Treatment with those inhibitors showed a reduction in soluble Sez6 (sSez6) levels to 27% (elenbecestat, Biogen, Eisai Co., Ltd., Tokyo, Japan), 17% (Shionogi compound **1**) and 39% (Shionogi compound **2**), compared to animals fed with vehicle pellets. We observed a significant decrease in the number of dendritic spines with Shionogi compound **1** after 21 days of treatment but not with Shionogi compound **2** or with elenbecestat, which did not show cognitive worsening in clinical trials. In conclusion, highly selective BACE1 inhibitors do alter dendritic spine density similar to non-selective inhibitors if soluble (sSez6) levels drop too much. Low-dose BACE1 inhibition might be reasonable if dosing is carefully adjusted to the amount of Sez6 cleavage, which can be easily monitored during the first week of treatment.

## 1. Introduction

Alzheimer’s disease (AD) is the most prevalent form of dementia globally and the amount of affected people is expected to rise dramatically as a consequence of the increased growth and aging of the population. A characteristic hallmark of AD is the massive deposition of the amyloid-beta peptide (Aβ) and ultimately the formation of Aβ plaques in the brain. The amyloid cascade hypothesis states that the accumulation of Aβ is an early pathogenic event in the development of AD [1,2]. Beta-site amyloid precursor protein cleaving enzyme 1 (BACE1) is considered as the rate-limiting enzyme for the production of Aβ. Therefore, the inhibition of BACE1 activity is considered a potential drug target against AD [3,4]. However, to date, all clinical trials using BACE1 inhibitors have been discontinued due to unfavorable side effects or a lack of efficacy. Studies reported adverse effects such as cognitive worsening, brain atrophy, or other neuropsychiatric effects [5,6,7]. Our own preclinical studies have revealed altered dendritic spine plasticity with various BACE inhibitors including clinical compounds [8,9,10]. Most BACE inhibitors used in clinical trials or preclinical studies lack discrimination between BACE1 and its homolog BACE2, which could raise new potential risks for patients.

BACE2 is mainly expressed in peripheral tissues but is also found in neuronal subtypes and glial cells in the mouse brain [11,12]. Some known functions of BACE2 include glucose homeostasis and hair pigmentation [13]. For example, BACE2 mediates the cleavage of PMEL in pigment cells, and the inhibition of BACE2 activity results in hair depigmentation in mice treated with BACE inhibitors [14,15]. Yet, the exact role of BACE2 in the brain is still poorly understood. Thus, BACE inhibition may cause some side effects due to the cross-inhibition of BACE2.

Therefore, in this study, we evaluated the impact of two new and highly BACE1 selective inhibitors in comparison with a slightly BACE1 selective inhibitor that was already in the clinic [16] and did not show cognitive worsening (elenbecestat) on dendritic spine dynamics and synaptic plasticity in mice. To clarify the affinity of the used inhibitors against BACE1, we first treated adult mice with pelleted compounds Shionogi **1**, Shionogi **2**, elenbecestat, or vehicle. Our results confirmed the preferential selectivity for BACE1 over BACE2, and we could observe a remarkable decrease in Aβ levels while BACE2 substrate levels remained unchanged. In order to assess the effect of selective BACE1 inhibition on dendritic spine plasticity, we performed longitudinal two-photon in vivo imaging in the cortex of transgenic mice expressing eGFP in a subset of pyramidal layer V neurons. By investigating structural spine plasticity in vivo, we could show that especially the Shionogi **2** compound, similar to elenbecestat, did not affect dendritic spine plasticity and spine turnover over time in mice treated over 21 days. However, Shionogi compound **1** did significantly reduce dendritic spine density, similar to that observed with non-selective BACE inhibitors like verubecestat [17].

## 2. Results

### 2.1. IC50 of BACE1 and BACE2 in Biochemical Enzymatic Assays

The IC50 of elenbecestat, Shionogi **1**, and Shionogi **2** against BACE1 was 3.9 nM, 3.9 nM, and 7.7 nM, respectively (Table 1). The IC50 of elenbecestat, Shionogi **1**, and Shionogi **2** against BACE2 was 46 nM, 148 nM, and 307 nM, respectively. These results indicate that all three inhibitors exhibit a strong selectivity for BACE1 over BACE2. Within 3 weeks of treatment, these compounds did not induce hair depigmentation.

### 2.2. BACE Inhibition Reduces Aβ and sSez6 Levels

To confirm that the novel BACE1 inhibitors have inhibitory activity toward BACE1, the brain levels of Aβ and soluble Seizure protein 6 (sSez6), which are cleavage products of BACE1 [18,19], were measured in the soluble brain fraction. All compounds reduced brain Aβ and sSez6 levels: elenbecestat (Aβ level reduction to 46%, sSez6 level reduction to 27% compared to vehicle), Shionogi compound **1** (Aβ level to 32%, sSez6 level to 17% compared to vehicle), and Shionogi compound **2** (Aβ level to 67%, sSez6 level to 39% compared to vehicle) (Figure 1). Brain Aβ and sSez6 were constantly reduced throughout the day.

### 2.3. Shionogi **2** Treatment Does Not Alter Dendritic Spine Plasticity

Recent clinical trials using different BACE inhibitors to treat AD have been terminated due to unfavorable side effects, possibly emerging from the non-discrimination of used inhibitors between BACE1 and BACE2 [20,21]. To investigate whether selective BACE1 inhibition causes alterations in synaptic spine plasticity, we treated adult GFP-M mice with BACE1-preferring inhibitors Shionogi **1** or Shionogi **2** and monitored in vivo dendritic spine dynamics in the apical dendritic segments of layer V pyramidal neurons (Figure 2A). In addition to the control group treated with vehicle, we compared our results also to a group treated with elenbecestat. We did not observe a significant difference in dendritic spine density in mice treated with Shionogi **2**, compared to the other groups (*p* = 0.1711) (Figure 2B). However, we observed a significant decline in total spine density after 21 days of treatment with Shionogi **1**, compared to vehicle (** *p* = 0.0093), elenbecestat (*^#^ p* = 0.0305), or Shionogi **2** (*^+^ p* = 0.0111). Next, we analyzed spine formation and elimination dynamics. We did not observe any significant differences in the elimination (*p* = 0.3857) and formation (*p* = 0.9183) of dendritic spines between treatments (Figure 2C,D). Moreover, the density of stable spines, as well as the density of transient spines, was not significantly affected (Figure 2E,F). Taken together, these results suggest that our clinical trial candidate Shionogi **2** does not have any effect on dendritic spine plasticity, neither in the formation nor in the elimination of spines in the cortical neurons of mice.

## 3. Discussion

BACE inhibition is a widely studied therapeutic approach to treat AD. As BACE1 is the responsible enzyme for the production of Aβ, research is focused on finding a treatment to lower Aβ levels by inhibiting BACE activity. However, clinical trials using non-selective inhibitors that do not discriminate between BACE1 and its homolog BACE2, like verubecesat, were discontinued. Cognitive worsening within the first months of treatment was one of the most problematic side effects that resulted in the termination of those trials. In preclinical studies using various non-selective BACE1 inhibitors including verubecesat, we observed a significant decrease in dendritic spine plasticity in the cerebral cortex within 2 weeks of treatment [17]. This decrease was reversible and seen also in other brain regions like the hippocampus. However, one of the very few clinical studies that do not report cognitive worsening used a compound that had a selectivity for BACE1 (elenbecestat). We therefore asked whether synaptic side effects may in part arise from the cross-inhibition of BACE2 [22]. Therefore, we evaluated the effect of elenbecestat and two newly developed, highly BACE1 selective inhibitors in a preclinical mouse study. We investigated the effect on dendritic spine plasticity by using longitudinal in vivo two-photon imaging of spines in the apical dendritic tufts of pyramidal V neurons in the cerebral cortex. The results of our current study show that elenbecestat and our candidate compound Shionogi **2** do not alter dendritic spine density in GFP-M mice (Figure 2). In detail, the density of gained and lost spines does not change under the treatment. However, different from Shionogi **2** and elenbecestat, the compound Shionogi **1** significantly affected spine density after 21 days of treatment. The most likely explanation is that Shionogi **1** does affect Sez6 cleavage more strongly than elenbecestat or Shionogi compound **2**. Shionogi **1** was found to reduce sSez6 levels to 17% of the baseline levels of vehicle-fed mice, whereas compound Shionogi **2** and elenbecestat reduced sSez6 levels only to 39% and 27%. As shown previously, a strong reduction in sSez6 levels under BACE1 inhibition is most likely responsible for the spine density reduction we saw in the cerebral cortex and in the hippocampus in mice [8,9,10]. Based on the results from the present study, we conclude that a reduction in sSez6 to 17% of baseline levels is critical for dendritic spine plasticity. However, above 27%, it is safe as we did not observe spine loss during treatment with elenbecestat or Shionogi **2**. Since the BACE1 selectivity of Shionogi compounds **1** and **2** is very similar, we assume that spine loss is a consequence of a strong BACE1 inhibition and not due to a strong selectivity for BACE1 over BACE2. 

In conclusion, Shionogi compound **2** might be a potential new BACE inhibitor for a long-term low-dose therapy following Aβ immunization. An individual adjustment of dosing could potentially be based on measuring soluble Sez6 levels in body fluids after a few days of treatment.

## 4. Material and Methods

### 4.1. BACE1 Inhibitors

Elenbecestat and Shionogi 1 and 2 were synthesized and provided by Shionogi & Co., Ltd., Osaka, Japan. Mice were fed ad libitum with food pellets containing either BACE inhibitor or vehicle for 21 days. The inhibitors were formulated as follows: elenbecestat (20 mg/kg food, KLIBA), Shionogi **1** (15 mg/kg food), Shionogi **2** (25 mg/kg food) (Research Diets Inc., New Brunswick, NJ, USA).

### 4.2. Animals

Male 2.5-month-old GFP-M mice (Tg(Thy1-EGFP)MJrs/J, Stock #007788, Jackson Laboratory, Bar Harbor, ME, USA) were used for in vivo spine imaging [23]. All animals were bred under pathogen-free conditions in the animal housing facility at the Center for Neuropathology and Prion Research of Ludwig Maximilian University Munich and were provided ad libitum with food pellets and water. Mice were group-housed until cranial window implantation, after which they were housed separately. The body weight and food intake were measured weekly during the treatment period with BACE inhibitor. All applicable international, national, and/or institutional guidelines for the care and use of animals were followed. The in vivo spine imaging experiments were conducted in accordance with the National Guidelines for Animal Protection and approved by Ludwig Maximilian University and the government of Upper Bavaria, Germany (Az. 55.2-1-54-2532-214-2016).

### 4.3. Biochemical Enzymatic Assays

BACE1 enzymatic activity was determined in a homogeneous time-resolved fluorescence (HTRF) assay, and BACE2 enzymatic activity was determined in an enzyme-linked immunosorbent assay (ELISA)-based assay using an APP-derived peptide that contains the K670N + M671L (Swedish) double mutation of the APP β secretase cleavage site as a substrate. The recombinant human BACE1 (R&D Systems, MN, USA) was incubated (2 h at 25 °C) with the substrate and the inhibitor (9 or 10 concentrations at 1/5 steps from 10 μM). Fluorescence intensity was measured (excitation wavelength: 320 nm, measuring wavelength: 620 nm and 665 nm) using RUBYstar (BMG LABTECH, Ortenberg, Germany). The recombinant human BACE2 ectodomain was incubated (1 h at 37 °C) with the substrate and the inhibitor (10 concentrations at 1/4 steps from 25 μM). The count of chemi-luminescence in each well was measured using ARVOTM MX 1420 Multilabel Counter (PerkinElmer, Shelton, CT, USA). The inhibitory potential of the compounds on the enzymatic activity of BACE1 and BACE2 was determined by performing concentration–response curves.

### 4.4. Brain Sample Preparation and Measurement of Aβ Levels

C57BL/6 mice (CLEA Japan Inc., Tokyo, Japan) were fed with food pellets containing 0.02% elenbecestat, 0.0125% Shionogi compound **1**, and 0.025% Shionogi compound **2** for one week (performed at AAALAC compatible facilities). Mouse cerebral hemispheres were removed and frozen immediately with liquid nitrogen. The frozen cerebral hemispheres were transferred to a homogenized tube containing ceramic beads in an 8-fold volume of the weight of an extraction buffer (containing 0.4% DEA (diethylamine), 50 mmol/L NaCl, completed with protease inhibitor (Roche, Basel, Switzerland)) and incubated on ice for 20 min. Thereafter, the hemispheres were homogenized using MP BIO FastPrep-24 (MP-Biomedicals, Irvine, CA, USA) with Lysing matrix D 1.4 mm ceramic beads (20 s at 6 m/s). After spinning for 1 min, the supernatant was transferred to a centrifugation tube and centrifuged at 221,000× *g*, 4 °C, for 50 min. After centrifugation, the supernatant was used for measuring the brain Aβ40. The brain Aβ40 was measured using Human/Rat Amyloid beta 40 ELISA Kit (FUJIFILM Wako Pure Chemical Corporation, Osaka, Japan). The assay procedure was conducted according to instructions.

### 4.5. Measurement of Soluble Sez6 Levels in the Mouse Brain

The brain samples were heated at 95 °C in 1% SDS sample buffer with 100 mM DTT for 5 min, separated by SDS-PAGE on 4–20% precast polyacrylamide gels (Biorad, Feldkirchen, Germany), and transferred to PVDF membranes (Biorad, Feldkirchen, Germany). The membranes were blocked in 5% skim milk (Becton Dickinson, Heidelberg, Germany) in PBST (FUJIFILM Wako Pure Chemical Corporation, Osaka, Japan) and incubated with anti-Sez6 antibody [24] or anti-Actin antibody (Merck Millipore, Darmstadt, Germany) at 4 °C overnight. The membranes were washed with PBST three times and incubated with HRP-conjugated secondary antibodies (Jackson Immunoresearch, West Grove, PA, USA) at room temperature for 1 h. After final washes with PBST, the membranes were incubated with ECL prime for a few minutes, and signal was detected with LAS-3000 (FUJIFILM Wako Pure Chemical Corporation, Osaka, Japan). Densitometry quantification of Western blot signal was conducted using Multi Gauge V3.1 software (FUJIFILM Wako Pure Chemical Corporation, Osaka, Japan).

### 4.6. Cranial Window Implantation

Cranial window surgery was performed as previously described [25]. In brief, mice were deeply anesthetized with a combination of Ketamine (0.13 mg/g body weight, bela-pharm GmbH & co.KG, Vechta, Germany) and Xylazine (0.01 mg/g body weight, Bayer, Leverkusen, Germany) i.p. and placed in a stereotaxic frame (WPI). A circular piece of skull (4 mm) was removed over the somatosensory cortex, and a cover glass of the same diameter was attached to the skull opening. In addition, a custom-made metal z-bar was attached to the front of the skull for head repositioning during later two-photon imaging sessions. After surgery, mice were placed on a heating pad until they recovered from anesthesia. To prevent postoperative infections and pain, Enrofloxacin (Baytril, 5 mg/kg body weight, Bayer, Leverkusen, Germany) and Carprofen (Rimadyl, 4 mg/kg body weight, Zoetis, Berlin, Germany) were administered subcutaneously.

### 4.7. Longitudinal In Vivo Two-Photon Spine Imaging

After a post-surgery recovery period of 4 weeks, mice were imaged once per week over 5 consecutive weeks. For imaging, mice were anesthetized with 5% Isofluran (Cp Pharma, Burgdorf, Germany) in 95% O_2_ and 5% CO_2_. Anesthesia depth was then maintained with 1.5% Isoflurane throughout the whole imaging session which did not exceed 60 min. GFP-positive apical dendritic tufts of layer V pyramidal neurons were imaged with an LSM 7MP two-photon microscope (Zeiss, Oberkochen, Germany) equipped with a water immersion objective (20×, NA = 1.0, Zeiss) and a femtosecond laser (Mai Tai DeepSee, Newport Spectra Physics, Darmstadt, Germany) tuned to 880 nm. In total, three regions of interest (ROI) and up to ten GFP-positive dendrites per region were localized for image acquisition. The same ROI positions were retrieved for each imaging session by using specific XY coordinates. First, a z-stack overview image (512 × 512 µm, 0.8 µm/pixel) was acquired for each ROI. Then, high-resolution images (256 × 198 µm, 0.1 µm/pixel) of each dendritic segment with emerging spines were acquired with an axial resolution of 1 µm. Based on the specific XY coordinates of the overview z-stacks and an overlay grid, the same dendritic segments could be retrieved for each imaging time point.

### 4.8. Dendritic Spine Data Analysis

Plasticity of dendritic spines was analyzed as described before [26,27,28]. In total, 7–10 dendrites per animal with a length of ~15–60 µm were obtained for analysis. Spines of all shapes with a length of >0.4 µm were counted manually using Zen 2011 Black Edition (Zeiss, Oberkochen, Germany). Due to resolution limitations, spines emerging laterally from the shaft (xy-plane) but not from the z-plane were taken into consideration. Spines that did not change their position (<0.5 µm) on the dendritic segment between two imaging time points (one week) were counted as stable. Spines were counted as lost when their length was less than 0.4 µm and gained when their length was greater than 0.4 µm. In current study, mice were imaged once per week; therefore, we counted spines as transient when they appeared at one imaging time point (>0.4 µm length) but were lost at the next time point (<0.4 µm length) one week later, as already used previously [17]. The spine turnover rate was calculated as follows: (N_gained_ + N_lost_)/(2 × N_total_)/I_T_. N_gained_ represents the number of newly gained spines at one time point, N_lost_ is the number of depleted spines, N_total_ is the number of all spines at one time point, and I_T_ is the days between two imaging sessions [26,28].

### 4.9. Experimental Timeline

Until the first two imaging sessions, mice were fed with standard food pellets (Ssniff, Soest, Germany). Immediately after the second imaging time point, the pellets were exchanged by either BACE1 inhibitor or vehicle feed for the following three weeks.

### 4.10. Statistics

All statistical analyses were performed with GraphPad Prism 7 (GraphPad Software Inc., Boston, MA, USA). A two-way ANOVA with Bonferroni post hoc test was performed to analyze spine dynamics over time. A *p* value of <0.05 was defined to be statistically significant. The data present the mean ± SEM.

## Figures and Tables

**Figure 1 ijms-24-12283-f001:**
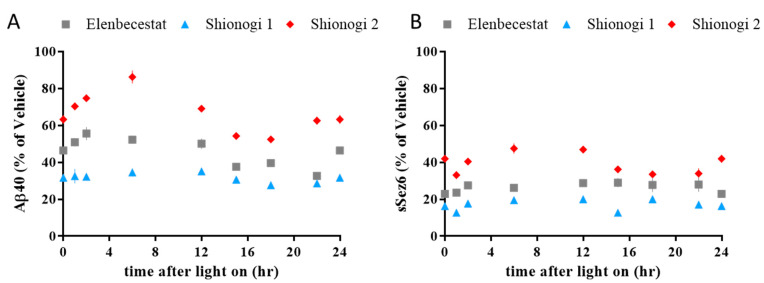
Brain Aβ and brain sSez6 levels throughout the day. BACE1 inhibitor treatment constantly reduced brain Aβ (**A**) and sSez6 (**B**) levels throughout the day, compared to vehicle: elenbecestat (Aβ level reduction to 46%, sSez6 level reduction to 27%), Shionogi compound **1** (Aβ to 32%, sSez6 to 17%), and Shionogi compound **2** (Aβ to 67%, sSez6 to 39%).

**Figure 2 ijms-24-12283-f002:**
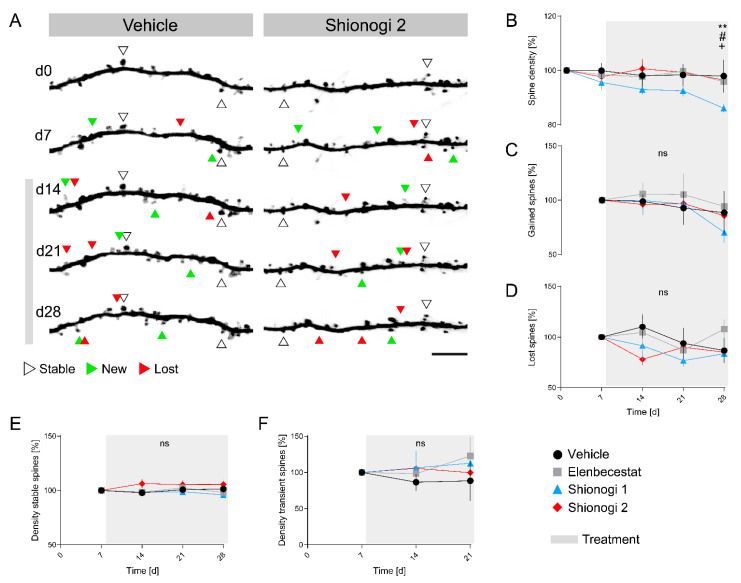
Effect of BACE inhibition with highly selective BACE1 inhibitors on dendritic spine density and plasticity. (**A**) Representative in vivo two-photon recordings of eGFP-labeled dendrites in cortical layer 5 neurons in GFP-M mice treated with vehicle or Shionogi **2** compound for four consecutive weeks. Arrowheads mark representative spines that were stable (white), newly formed (green), or lost (red). Gained spines that did not stabilize (green/red, present < 7 days) were defined as transient, whereas gained spines that did stabilize were defined as persistent (green/white, present > 7 days). Scale bar: 2 µm. (**B**) Dendritic spine density. (**C**) Density of gained spines. (**D**) Density of lost spines. (**E**) Density of stable spines. (**F**) Density of transient spines. N = 4–7 animals per group, n = 7–10 dendrites per animal. Data are presented as mean ± SEM. Bonferroni post hoc test: ^+^/^#^ *p* < 0.05, ** *p* < 0.01, from two-way ANOVA.

**Table 1 ijms-24-12283-t001:** Inhibitory activity of BACE inhibitors toward BACE1 and BACE2.

Biochemical Enzymatic Assay	Elenbecestat	Shionogi 1	Shionogi 2
BACE1/APP (IC50 nM)	3.9	3.9	7.7
BACE2/APP (IC50 nM)	46	148	307
BACE1/BACE2 (fold)	12	38	40

## Data Availability

Not applicable.

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
