# Peer review of "New Highly Selective BACE1 Inhibitors and Their Effects on Dendritic Spine Density In Vivo"

_ijms, 2023, doi:10.3390/ijms241512283_

Round 1

Reviewer 1 Report

In this short and rather preliminary paper, Pratsch et al. compared the effects of three BACE1 inhibitors with different degrees of selectivity for BACE1 over BACE2, using longitudinal in vivo optical recordings of dendritic spine dynamics over a period of 3 – 4 weeks. In addition, the authors determined IC50-values of the three compounds for BACE1 and BACE2, and measured their effects on Abeta and Sez6 levels.

The Herms lab is internationally renowned for their seminal work in the BACE field, combining state-of-the-art spine recordings in vivo with electrophysiological measurements and behavioral tests. It is all the more unexpected to get a manuscript for review that falls well below the excellent standard that characterizes publications from this lab. I regret to say that, in its present form, the manuscript is of rather low quality, the findings are not well explained and presented, making them difficult to understand in part, a discussion of the limitations of the study is lacking, and the conclusions are not entirely convincing.

Specific comments

(1) Although two compounds, Shionogi 1 and Shionogi 2, exhibit almost identical high selectivity for

BACE1, their effects on Abeta levels and spine density are vastly different, raising concerns that BACE1 selectivity is not the major mechanism preserving spine integrity, the more so since elenbecestat, which is much less selective for BACE1, performs much better in this paradigm than the highly BACE1 selective compound, Shionogi 1.

(2) Fig. 2, which presents the core findings of the paper, suffers from several shortcomings. In panel A, arrowheads are in part not unambiguously assigned to individual spines. Throughout panels B –E, data points are depicted without error bars and significance levels.

(3) In the Introduction, the authors mention that the three compounds did not affect BACE2 substrate levels, but I could not find experimental evidence for this notion in the Result section. Along these lines, did the compounds alter skin hair pigmentation?

(4) The title overstates the significance of the findings, since the authors measured dendritic spine dynamics, but not synaptic plasticity, which I would define as a change in synaptic efficacy in response to a plasticity-inducing stimulus. Please change the title accordingly.

 (5) In Fig. 1, the experimental protocol (times after light on) needs more explanation. Are data points representing single measurement? Reduction of Abeta and Sez6 to x % or by x %?

Author Response

Reviewer #1: Comments to authors:

(1) Although two compounds, Shionogi 1 and Shionogi 2, exhibit almost identical high selectivity for BACE1, their effects on Abeta levels and spine density are vastly different, raising concerns that BACE1 selectivity is not the major mechanism preserving spine integrity, the more so since elenbecestat, which is much less selective for BACE1, performs much better in this paradigm than the highly BACE1 selective compound, Shionogi 1.

Response: Thank you for calling attention to this point. We now included this important point in the discussion: “The results of our current study show that elenbecestat and our candidate compound Shionogi 2 does not alter dendritic spine density in GFP-M mice. In detail, the density of gained and lost spines does not change under the treatment. However, different to Shionogi 2 and elenbecestat, the compound Shionogi 1 significantly affected spine density after 21 days of treatment. The most likely explanation is that Shionogi 1 does affect Sez6 cleavage more strongly than elenbecestat or Shionogi compound 2. Shionogi 1 was found to reduce soluble Sez6 levels to 17% of baseline levels of vehicle fed mice, whereas compound Shionogi 2 and elenbecestat reduced soluble sez6 levels only to 39% and 27%.”

(2) Fig. 2, which presents the core findings of the paper, suffers from several shortcomings. In panel A, arrowheads are in part not unambiguously assigned to individual spines. Throughout panels B –E, data points are depicted without error bars and significance levels.

Response: We are very sorry. We now added error bars and significance levels in figure 1 and 2.

(3) In the Introduction, the authors mention that the three compounds did not affect BACE2 substrate levels, but I could not find experimental evidence for this notion in the Result section. Along these lines, did the compounds alter skin hair pigmentation?

Response: We included a sentence in the result section that we did not observed hair deigmentation with Shionogi 1 and 2.

(4) The title overstates the significance of the findings, since the authors measured dendritic spine dynamics, but not synaptic plasticity, which I would define as a change in synaptic efficacy in response to a plasticity-inducing stimulus. Please change the title accordingly.

Response: We changed the title to: New highly selective BACE1 inhibitors and their effects on dendritic spine density in vivo.

 (5) In Fig. 1, the experimental protocol (times after light on) needs more explanation. Are data points representing single measurement? Reduction of Abeta and Sez6 to x % or by x %?

Response: We changed the legend: BACE1inhibitor treatment constantly reduced brain Aβ and sSez6 levels throughout the day, compared to vehicle: Elenbecestat (Aβ level reduction to 46%, sSez6 level reduction to 27%), Shionogi compound 1 (Aβ to 32%, Sez6 to 17%), and Shionogi compound 2 (Aβ to 67%, Sez6 to 39%).

Reviewer 2 Report

In this communication paper, entitled "Novel BACE1 selective inhibitor does not affect synaptic 2 plasticity in vivo", Of Pratsch et al., they study the effect of a newly developed BACE inhibitors with a high selectivity for BACE1 (Shionogi 1 or 2). By using longitudinal in vivo two-photon microscopy, they investigated the effect of this inhibitor on dendritic spine dynamics of pyramidal neurons in the somatosensory cortex in mice. They find that BACE1 selective inhibitor markedly reduced brain Aβ and seems to show no adverse effects on dendritic spine plasticity. They suggest that selective BACE1 inhibition might provide an effective treatment to reduce the production of amyloid without cognitive side effects which may be caused by the inhibition of BACE2.

I am pleased for having had the opportunity to read this paper and I find very interesting their findings regarding the possibility to use specific BACE inhibitors without side effects and I hope there are future developments in the study of this inhibitor.

Author Response

Reviewer #2: Comments to authors:

I am pleased for having had the opportunity to read this paper and I find very interesting their findings regarding the possibility to use specific BACE inhibitors without side effects and I hope there are future developments in the study of this inhibitor.

Response: We thank this referee for his/her highly encouraging and positive remarks and greatly appreciate his/her interest in our manuscript.

Round 2

Reviewer 1 Report

The authors have thoroughly revised the manuscript, addressing my concerns in a very satisfying fashion. Moreover, by highlighting the critical role of Sez6 cleavage as a candidate predictor of the therapeutic outcome when administering BACE1 inhibitors, the manuscript has gained considerable momentum and should be of broad interest to the AD community.